# Novel RNA Viruses Discovered in Weeds in Rice Fields

**DOI:** 10.3390/v14112489

**Published:** 2022-11-10

**Authors:** Shufen Chao, Haoran Wang, Shu Zhang, Guoqing Chen, Chonghui Mao, Yang Hu, Fengquan Yu, Shuo Wang, Liang Lv, Long Chen, Guozhong Feng

**Affiliations:** 1State Key Laboratory of Rice Biology, China National Rice Research Institute, Chinese Academy of Agricultural Sciences, Hangzhou 311400, China; 2Institute of Plant Protection & Soil Fertilizer, Hubei Academy of Agricultural Sciences, Wuhan 430000, China; 3Institute of Plant Protection, Guizhou Academy of Agricultural Sciences, Guiyang 550000, China; 4Institute of Plant Protection, Liaoning Academy of Agricultural Sciences, Shenyang 110000, China; 5Sanya Agricultural Technology Extension and Service Centre, Sanya 572000, China

**Keywords:** weed, metatranscriptomics, RNA viruses, viral diversity, plant viruses

## Abstract

Weeds often grow alongside crop plants. In addition to competing with crops for nutrients, water and space, weeds host insect vectors or act as reservoirs for viral diversity. However, little is known about viruses infecting rice weeds. In this work, we used metatranscriptomic deep sequencing to identify RNA viruses from 29 weed samples representing 23 weed species. A total of 224 RNA viruses were identified: 39 newly identified viruses are sufficiently divergent to comprise new families and genera. The newly identified RNA viruses clustered within 18 viral families. Of the identified viruses, 196 are positive-sense single-stranded RNA viruses, 24 are negative-sense single-stranded RNA viruses and 4 are double-stranded RNA viruses. We found that some novel RNA viruses clustered within the families or genera of several plant virus species and have the potential to infect plants. Collectively, these results expand our understanding of viral diversity in rice weeds. Our work will contribute to developing effective strategies with which to manage the spread and epidemiology of plant viruses.

## 1. Introduction

Crop plants often grow alongside weeds and weed infestation is a worldwide challenge for crop production. Crops can be affected by weeds at any stage of growth. In addition to competing with crops for nutrients, water and space, weeds can serve as ‘bridge’ hosts of viruses, resulting in viral transmission in crop fields. These wild plants constitute potential reservoirs for viruses that may spread into crop plants, thereby resulting in epidemics or the emergence of novel viruses [1,2,3,4]. Many viruses have wide host ranges, comprising both wild plants and crop plants [5,6]. Viruses can emerge in crop plants from wild hosts, or conversely, virus infections in cultivated plants may spill over and affect wild plant populations [2]. Viruses in wild plants have been reported to be diverse and often do not exhibit evident symptoms [7,8,9]. In rice fields, various weeds grow with rice plants. However, little is known about the viruses originating from weeds in rice fields. 

Metatranscriptomics has proven to be a powerful approach by which to uncover hidden viruses without any prior knowledge. It provides sufficient coverage to reconstruct complete or near-complete viral genomes and allows straightforward characterization of viral diversity. To date, metatranscriptomics has been used to explore viral diversity in respect of infected hosts, including humans, arthropods and plants [10,11,12]. Using metatranscriptomic approach, 1445 RNA viruses were discovered from over 200 invertebrate species sampled across nine animal phyla [13]. A total of 13 RNA viruses were detected from sterile rice plants [14]. Metatranscriptomics has advantages in terms of characterizing RNA viromes: (i) it can uncover entire viromes, including those from coinfected pathogens; (ii) it provides reliable quantification and assessment of both viral and host RNAs; (iii) it is relatively simple, requiring minimal sample processing [15]. 

In this study, we characterized the viromes of 29 weed samples covering 23 weed species collected from rice fields in 4 different rice production regions in China. A comprehensive analysis of metatranscriptomic sequencing revealed hundreds of known and novel viruses detected in the weed samples. We also determined the type, number and abundance for each virus within the weed samples. The results provide insight into virus diversity and evolution and offer valuable information with which to facilitate the development of efficient strategies for weed management.

## 2. Materials and Methods

### 2.1. Sample Collection

Samples were collected from Hangzhou, Xiaogan, Guiyang and Sanya in 2018 and 2019. A total of 29 weed samples belonging to 23 weed species were obtained from rice fields. These samples were from two classes: Dicotyledoneae and Monocotyledoneae. (i) For Dicotyledoneae, we sampled 11 species within the orders Euphorbiales, Campanulales, Tubiflorae, Centrospermae, Plantaginales, Ranales and Rosales; (ii) For Monocotyledoneae, we sampled 12 species within the orders Graminales, Helobiae, Cyperales and Farinosae (Appendix A). Samples were stored at −80 °C for later RNA extraction.

### 2.2. RNA Library Construction and Sequencing

The total RNA of the leaf and stem of the samples was extracted using TRIzol LS reagent (Invitrogen, Carlsbad, CA, USA) and the extracted RNA was treated with DNase. The qualitative and quantitative methods for total RNA extraction were as follows: (i) RNA samples were firstly qualitatively analyzed by 1% agarose gel electrophoresis for possible contamination and degradation; (ii) the purity and concentration of RNA were then examined by NanoPhotometer^®^ spectrophotometer; (iii) RNA samples were accurately confirmed using a Qubit2.0 Fluorometer; (iv) finally, RNA integrity and quantity were measured using an RNA Nano 6000 Assay Kit with a Bioanalyzer 2100 system.

One library contains a single individual from one species at the same sampling location. In total, 29 libraries were constructed from 29 weed samples. RNA libraries were constructed using the TruSeq total RNA library preparation protocol (Illumina, San Diego, CA, USA). Briefly, the ribosomal RNA was depleted from total RNA using a Ribo-Zero Magnetic kit (Plant Leaf) (Illumina, CA, USA) following the manufacturer’s instructions. The remaining RNA was then fragmented into 250–300 bp fragments and then reverse-transcribed into cDNA. The remaining overhangs of double-stranded cDNA were converted to blunt ends by exonuclease polymerase activities. After the 3’ end of the DNA fragment was adenylated, the sequencing adaptors were ligated to the cDNA. In order to preferentially select cDNA fragments of 250 to 300 bp in length, the library fragments were purified using the AMPure XP system (SPRI beads) (Beckman Coulter, Brea, CA, USA). Amplification of cDNA was performed using PCR. Pair-end (150 bp) sequencing of the RNA library was performed using the HiSeq 2500 platform (Illumina, Sandiego, CA). All library preparation and sequencing steps were performed by Novogene Co., Ltd. (Tianjin, China).

### 2.3. RNA Virus Discovery

Sequencing reads from all 29 samples were trimmed and assembled individually de novo using Trinity [16]. To maximize sensitivity and minimize false-positive results, the assembled contigs were compared against all reference RNA virus proteins downloaded from Genbank by using BLASTx at an E-value ≤ 10^−4^. The resulting contigs were then blasted using non-redundant nucleotide (nt) and protein (nr) databases to remove non-viral sequences. The assembled contigs were compared to the Conserved Domain Database (CDD, version 3.16) with an E-value of 10^−2^ to detect highly divergent viruses. Quality-filtered viral contigs with unassembled overlaps were pooled using SeqMan in the Lasergene software package v7.1 (DNAStar). To confirm the assembly results, the reads were mapped to the viral genome using Bowtie2 [17] and checked for possible assembly errors using the Integrated Genomics Viewer (IGV) [18]. We used RSEM to count the reads mapped to the viral genomes [19], and the average mapping depths were computed as the total numbers of mapped bases divided by the viral length. To validate assembled sequences, we firstly used Poly(A) Tailing Kit (Applied Biosystems, Waltham, MA, USA) to add poly(A) to the 3′ end of the total RNA, then used SuperScriptTM III Reverse Transcription (ThermoFisher, Waltham, MA, USA) for reverse transcription. Finally, RT-PCR amplification was performed using PrimerSTAR Max DNA Polymerase (Takara, Kyoto, Japan). The amplified products were cloned into pGEM T-Easy vector (Promega, Madison, WI, USA) to Sanger sequencing. Genome termini were determined using 5′/3′-Full RACE Core Set with PrimeScript™ RTase (TaKaRa, Kusatsu City, Japan). To confirm the assembly results, reads were mapped to the putative viral genomes with Bowtie2 and checked using the Integrated Genomics Viewer.

### 2.4. Virus Genome Annotation and Phylogenetic Analysis

Viral genomes were annotated using the following methods: (i) for viruses with closely related relatives, the functional domains were predicted by homology with known viral proteins; (ii) for viruses without a homologue in a closely related virus, the functional domains with each ORF was predicted using a domain-based blast search against the conserved domain database (https://www.ncbi.nlm.nih.gov/Structure/cdd/wrpsb.cgi, accessed on 15 August 2021) (expected threshold of 10^−5^). Signal peptides were determined using the SignalP-5.0 web service (http://www.cbs.dtu.dk/services/SignalP/, accessed on 11 September 2021). Transmembrane helices in ORFs were predicted using TMHMM v.2.0.

For the viruses with multiple segments, the segments were identified by homology to the proteins of the related reference proteins. For the potential segments without homology, we checked: (i) the sequencing depth of the segments; (ii) the presence of conserved regulatory sequences in 5’ and/or 3’ non-coding regions of the genome; (iii) the segments were found in the same samples; (iv) the phylogenetic positions of related proteins [13]. 

To determine the phylogenetic relationships of RNA viruses identified in our study, all replicase proteins were collected to infer the evolutionary history. Amino acid sequences of viruses were aligned using the E-INS-i algorithm in MAFFT (version 7.490) [20], and the ambiguously aligned regions were then removed by trimAl (version 1.2) using automated1 code [21]. Phylogenetic trees were then inferred using the maximum likelihood (ML) approach implemented in IQ-TREE (version 1.6.12) with 1000 bootstrap replicates, using “MFP” to select the best model [22]. We also inspected these trees using ML and the Bayesian method in MrBayes 3.2.7a [23]. Similar results were generated using these two programs. Only maximum likelihood phylogenetic trees generated by IQ-TREE are shown. 

### 2.5. Estimation of Viral Transcript Abundance and Virus Names

To determine the abundance of RNA transcripts, the clean reads (non-rRNA reads) from each library were mapped to assembled transcripts and the total percentage of viral sequences in the library was calculated. An abundance of viruses that was greater than or equal to 0.01% of non-rRNA reads indicated high abundance; otherwise, it was low abundance. Based on these results, viruses discovered in this study were named according to the following regulations: (i) for viruses with high abundance (≥0.01% of non-rRNA reads), the virus name comprises the geographic region of sampling, the host name (such as *Aeschynomene indica*), the closest family and a virus number; (ii) for viruses with low abundance (<0.01% of non-rRNA reads), the virus name comprises the geographic region of sampling, the closest family and a virus number.

## 3. Results

To determine the viral diversity of weeds in rice fields, we collected 29 weed samples from four major regions of rice production, including Hangzhou (subtropical region), Xiaogan (subtropical region), Guiyang (plateau region) and Sanya (tropical region) (Appendix A). These comprised 23 weed species from the families Alismataceae, Solanaceae, Amaranthaceae, Compositae, Scrophulariaceae, Gramineae, Euphorbiaceae, Plantaginaceae, Ranunculaceae, Asteraceae, Cyperaceae, Pontederiaceae, Acanthaceae and Leguminosae, covering all major weed species in rice fields. Total RNA was extracted from the collected samples, and 29 RNA libraries for high-throughput sequencing were constructed from rRNA-depleted RNAs. We performed deep metatranscriptomic sequencing of weed samples. One library contains a single individual from one sample at the same sampling location. In total, 29 libraries were constructed comprising 29 rice weed samples (Appendix A). RNA sequencing resulted in 64 million to 116 million reads per pool and a total of 368.29 Gb of clean bases were obtained from 29 libraries. Genomes were then assembled de novo and screened for RNA viruses. 

In total, we identified 304 RNA viral genomes containing RNA-dependent RNA polymerase (RdRp) domain. Using a series of protein sequence similarity-based BLAST searches, 26 contigs matched closely to known viruses, and 278 contigs were novel and distinct from publicly available viruses in GenBank (Appendix A). The numbers of viruses identified in each library were significantly different. In Sanya, the weed *Leptochloa chinensis* harbors up to 99 viruses, while the weeds *Eclipta prostrata* and *Acalypha australis* in Hangzhou, *Echinochloa crus-galli*, *Alternanthera philoxeroides* and *Aster indicus* in Guiyang and *Justicia procumbens* in Xiaogan harbor only 1 virus (Appendix A).

Phylogenetic analysis showed that the 302 putative virus genomes belonged to 211 novel RNA viruses and 13 currently assigned RNA viruses (Appendix A). This suggests that a large numbers of RNA viruses are present in weeds. Among the identified RNA viruses, 196 viruses were positive-sense RNA viruses, 24 viruses were negative-sense RNA viruses and 4 viruses were double strand RNA viruses. The identified positive-sense RNA viruses clustered into 18 viral families, including *Astroviridae*, *Atkinsviridae*, *Botourmiaviridae*, *Closteroviridae*, *Dicistroviridae*, *Duinviridae*, *Fiersviridae*, *Fusariviridae*, *Marnaviridae*, *Mitoviridae*, *Narnaviridae*, *Nodaviridae*, *Solemoviridae*, *Solspiviridae*, *Steitzviridae*, *Tombusviridae*, *Tymoviridae* and *Virgaviridae*. The identified negative-sense RNA viruses grouped within the families *Peribunyaviridae*, *Phenuiviridae*, *Leishbuviridae*, *Alphabunyaviridae* and *Betabunyaviridae*. All the dsRNA viruses were clustered within the family *Partitiviridae* (Figure 1). Notably, 39 novel viruses are sufficiently divergent to comprise new families and genera (Figure 2).

### 3.1. Characterization of Postive-Sense RNA Viruses

A total of 196 ssRNA (+) viruses were identified from the collected weeds. Among these viruses, 12 belong to the phylum *Pisuviricota,* which includes the families *Dicistroviridae* (*n* = 5), *Solemoviridae* (*n* = 2), *Astroviridae* (*n* = 1) and *Marnaviridae* (*n* = 4). Twenty clustered within the Phylum *Kitrinoviricota,* which includes the families *Tombusviridae* (*n* = 16), *Nodaviridae* (*n* = 1), *Tymoviridae* (*n* = 1), *Closteroviridae* (*n* = 1) and *Virgaviridae* (*n* = 1), Eighty-five ssRNA(+) viruses belong to the families of mycoviruses (Appendix A), including *Botourmiaviridae* (*n* = 41), *Mitoviridae* (*n* = 26), *Narnaviridae* (*n* = 17) and *Fusaviridae* (*n* = 1). Seventy-nine are bacteriophages (*Leviviricetes*, *n* = 79) and belong to the Phylum *Lenarviricota* (Appendix A).

**(i) *Dicistroviruses***. The family *Dicistroviridae* includes three genera, *Aparavirus*, *Cripavirus* and *Triatovirus,* with positive-sense genomes ranging from 8 kb to 10 kb [24]. Here, four new dicistroviruses, Sanya dicistro-like virus (SDiV) 1 to 3 and Guizhou dicistro-like virus (GDiV) 1, were detected in weeds. These new dicistroviruses have genome lengths between 7275 and 9272 bp, and share 28.5% to 40.9% of their amino acid (aa) identities with the closely related viruses. In Sanya, SDiV1 was discovered in weed species *Leersia hexandra, Leptochloa chinensis*, *Eleocharis dulcis*, *Monochoria vaginalis* and *Cyperus difformis*; SDiV2 was identified in weed *Leersia hexandra*, and SDiV3 was identified in weeds *Cyperus difformis* and *Monochoria vaginalis*. However, GDiV1 was identified in the weed *Paspalum distichum* in Guiyang (Appendix A). In addition to these novel dicistroviruses, we also identified a variant of the already-described Beihai picorna-like virus 77 (BPlV77) (GenBank no. NC_032551.1) [13]. BPlV77-HN was annotated as a variant of BPlV77, and sequence similarity analysis showed that the predicted RdRp protein of BPlV77-HN was 99.8% identical to BPlV77 (Appendix A). Phylogenetic analysis showed that the four novel dicistroviruses were not classified into the well-defined genus in the family *Dicistroviridae* (Appendix A). SDiV1 and GDiV1 formed a separate branch, SDiV2 clustered within the Human blood-associated dicistrovirus clade, while SDiV3 was located between Shahe picorna-like virus 9 and Changjiang picorna-like virus 11, which were previously identified in invertebrate species [13]. 

**(ii) *Solemoviruses***. The family *Solemoviridae* includes four genera: *Sobemovirus*, *Polerovirus*, *Enamovirus* and *Polemovirus*. The genomes of the members contain 4 to 10 open reading frames (ORFs) with a positive-sense RNA molecule of 4 kb to 6 kb without a 3′-poly(A) tail [25]. The family includes important plant pathogens with high economic impact, with rice yellow mottle virus, potato leafroll virus and sugarcane yellow leaf virus (ScYLV) being some of the most devastating [26,27,28]. Here, we identified two novel solemoviruses: Guiyang Paspalum distichum solemo-like virus 1 (GPDV1) and Xiaogan Aeschynomene indica solemo-like virus 1 (XAISV1). GPDV1 has a genome of 5786 bp in size and was identified in the weed *Paspalum distichum* in Guiyang. XAISV1 has a genome of 5689 bp in size and was detected in the weed *Aeschynomene indica* in Xiaogan. GPDV1 and XAISV1 share 72.8% and 71.1% aa identities with the closely related viruses, respectively, and displayed a high abundance ranging from 0.030 to 3.916, suggesting that *Paspalum distichum* was the possible host of GPDV1, and *Aeschynomene indica* is a possible host of XAISV1. Phylogenetic analysis showed that GPDV1 and XAISV1were surrounded by members of the *Solemoviridae* family (Figure 3). XAISV1 clustered with alfalfa enamovirus-1 (AEV-1), pea enation mosaic virus 1 (PEMV-1) and grapevine enamovirus-1 (GEV-1), with short branch lengths and sharing high similarities (57% to 86% aa identities) (Figure 3 and Appendix A). However, GPDV1 was assigned into the genus *Polemovirus* and formed a small clade with ScYLV and wheat leaf yellowing-associated virus (WLYaV); GPDV1 shared 62.2% and 62.9% aa identities with ScYLV and WLYaV, respectively (Figure 3 and Appendix A). Importantly, plants infected by AEV-1, PEMV-1 and GEV-1 exhibited obvious symptoms such as leaf chlorosis or leaf stunting [29,30,31]. ScYLV and WLYaV can induce yellow leaf disease in sugarcane and wheat [28,32]. GPDV1 has a similar genome structure to the closely related viruses WLYaV and ScYLV; XAISV1 has a similar genome structure to the closely related viruses AEV-1, PEMV-1 and GEV-1. Sequence similarity analysis and phylogenetic results showed that the two novel solemoviruses may have the potential to infect weeds and cause viral diseases (Appendix A).

**(iii) *Astroviruses***. The family *Astroviridae* comprises two genera: *Mamastrovirus*, which infects mammals, and *Avastrovirus*, which infects birds. Many members of the family *Astroviridae* infect plants and caused various viral diseases [33,34]. Members of the family *Astroviridae* have genome lengths ranging from 6.8 to 7.9 kb [35]. We characterized Sanya astro-like virus 1 (SAlV1) as a novel species of the family *Astroviridae*. SAlV1 was discovered in the weed *Leersia hexandra* in Sanya and has a genome length of 4099 bp. This virus shared 57.6% aa identity with the closely related virus (Appendix A). Phylogenetic analysis showed that SAlV1 clustered with the viruses, Astroviridae sp. and Bufivirus UC1, but appeared to represent a separate evolutionary lineage (Appendix A). Phylogenetic analysis showed that the small branch comprising these four viruses may constitute a new genus of the family *Astroviridae*.

**(iv) *Marnaviruses***. The family *Marnaviridae* includes seven genera, *Salisharnavirus*, *Sogarnavirus*, *Bacillarnavirus*, *Locarnavirus*, *Labyrnavirus*, *Kusarnavirus* and *Marnavirus*, with positive-sense RNA genomes of 8.6 to 9.6 kb containing one or two ORFs [36]. We identified four viruses as novel members of the family *Marnaviridae*, Sanya marna-like virus (SMaLV) 1 to 3 and Xiaogan marna-like virus 1 (XMaLV1). They have genome lengths ranging from 5717 to 9184 bp and show 46.2% to 70.2% aa sequence similarity with the closely related viruses (Appendix A). SMaLV1 was identified in weeds *Leptochloa chinensis* and *Cyperus difformis* in Sanya, while XMaLV1 was discovered in the weeds *Bidens tripartita* and *Digitaria sanguinalis* in Xiaogan (Appendix A). The phylogenetic analysis showed that SMaLV1, SMaLV2 and XMaLV1 were assigned into the genus *Locarnavirus*, while XMaLV1 clustered within the genus *Labyrnavirus* (Appendix A). 

**(v) *Tombusviruses*.** The members of the family *Tombusviridae* are divided into three subfamilies, 16 genera and have non-segmented (monopartite) linear genomes 3.7 to 4.8 kb in size [37]. We identified 16 novel tombusviruses in 12 weed samples: Guiyang paspalum paspaloides tombus-like virus (GPpTV) 1 to 2, Guiyang tombus-like virus (GTV) 1 to 3, Hangzhou tombus-like virus (HTV) 1 to 2, Sanya leptochloa chinensis tombus-like virus 1 (SLCTV1) and Sanya tombus-like virus (STV) 1 to 8. These newly identified viruses have genome lengths ranging from 2591 to 4811 bp, and they share 31.6% to 80.1% aa identities with the closely related viruses (Appendix A). Notably, STV1 was detected in six weed samples in Sanya (Appendix A). HTV1 and HTV2 were identified in the weeds *Alternanthera philoxeroides* and *Acalypha australis* in Hangzhou, respectively. STV8 was identified in the weed *Leersia hexandra* in Sanya. In addition, four novel viruses (SLCTV1, STV2, STV3 and STV7) were identified in the weed *Leptochloa chinensis* in Sanya, and three novel viruses (STV4 to 6) were detected in the weed *Monochoria vaginalis* in Sanya. Both GPpTV1 and GPpTV2 were discovered in the weed *Paspalum distichum* in Guiyang, while GTV1 to 3 were discovered in the weeds *Ranunculus japonicus*, *Paspalum thunbergii* and *Plantago depressa* in Guiyang, respectively (Appendix A). In addition, STV1, SLCTV1, GPpTV1 and GPpTV2 displayed a high abundance ranging from 0.012 to 0.139, suggesting that *Leptochloa chinensis* was a possible host of STV1 and SLCTV1, and weed *Paspalum distichum* was possible host of GPpTV1 and GPpTV2. Phylogenetic analysis showed that GTV1, GTV2, STV1 to -3 and SLCTV1 were assigned into the *Procedovirinae* subfamily, GPpTV1 and HTV2 clustered within the genus *Umbravirus*, GPpTV2 clustered within the genus *Dianthovirus* and HTV1 was located outside the genus *Umbravirus*. The other six tombusviruses (STV4 to -8 and GTV3) clustered within the previously described tombusviruses from invertebrate species by high-throughput sequencing. Importantly, SLCTV1 was closely related to oat chlorotic stunt virus (OCSV) (GenBank no. NC_003633), the only member of the genus *Avenavirus*, and SLCTV1 shared 58.7% aa identity with OCSV (Figure 4 and Appendix A). GPpTV2 clustered within the subfamily *Regressovirinae* containing Carnation ringspot virus (CRSV), red clover necrotic mosaic virus (RCNMV) and sweet clover necrotic mosaic virus (SCNMV), and they shared high aa sequence similarity (61.0% to 63.1% aa identity) (Figure 4 and Appendix A). SLCTV1 has a similar genome structure to the closely related viruses OCSV, and GPpTV2 has a similar genome structure to the closely related viruses CRSV, RCNMV and SCNMV (Appendix A). Previous studies have shown that CRSV, RCNMV and SCNMV caused viral disease in host plants with different symptoms, such as leaf discoloration, ring spot and stunted plant growth [38]. Thus, SLCTV1 and GPpTV2 have the potential to infect cereal crops and cause diseases.

**(vi) *Nodaviruses***. Viruses in the family *Nodaviridae* are divided into two genera: *Alphanodavirus* and *Betanodavirus*. The genomes of the members were composed of two molecules of single-stranded positive-sense RNA: RNA1 (3.1 kb) encoding RdRp, and RNA2 (1.4 kb) encoding capsid protein [39]. We characterized Sanya noda-like virus (SNV) as a novel member of the family *Nodaviridae*. SNV was detected in three weeds, *Leptochloa chinensis*, *Monochoria vaginalis* and *Cyperus difformis* in Sanya, and it shared 54.5% aa identities with the closely related virus (Appendix A). The genome of SNV was 3843 bp in size. The phylogenetic analysis showed that SNV was located outside the members of the genera *Alphanodavirus* and *Betanodavirus*, while it formed a single cluster with Hubei noda-like virus 20 and Hubei noda-like virus 19 (Appendix A).

**(vii) *Tymoviruses***. The family *Tymoviridae* contains three genera: *Marafivirus*, *Maculavirus* and *Tymovirus*. Viruses in the family *Tymoviridae* have a wide range of hosts [40]. The genomes of the members of the genera *Marafivirus* and *Tymovirus* each contained a 16-nt conserved sequence at the 3’ end of ORF1, namely “marafibox” (CA(G/A)GGUGAAUUGCUUC) and “tymobox” (GAGUCUGAAUUGCUUC), respectively [41]. We characterized Guiyang Paspalum distichum tymo-like virus 1 (GPdTV1) and classified it as a novel member of the family *Tymoviridae*. GPdTV1 has genome length of 6725 bp, and shares 45.3% aa identity with the closest related virus (Appendix A). The genome of GPTV1 has a conserved sequence that is highly similar to “marafibox” (Appendix A). GPdTV1 has a high abundance of 0.019, suggesting that *Paspalum distichum* is a possible host of GPdTV1. The phylogenetic analysis showed that GPdTV1 was located between the genera *Marafivirus* and *Maculavirus*, and it was closely related to Citrus virus C (Appendix A). GPdTV1 has a genome structure closely matching Citrus virus C, and it contains a putative movement protein for virus infection [42]. Together, these results indicate that GPdTV1 is a new member of the genus *Marafivirus*.

**(viii) *Virgaviruses***. The family *Virgaviridae* includes seven genera. Viruses in the family *Virgaviridae* are plant viruses with rod-shaped virions, an ssRNA genome with a 3′-terminal tRNA-like structure and a replication protein typical of alpha-like viruses [43]. Here, we identified a novel virus, Hangzhou virga-like virus (HVlV) in the weed *Lindernia procumbens* acquired in Hangzhou. The genome size of HVlV was 2918 bp, and HVlV shared 53.4% aa identity with the closest related virus (Appendix A). Phylogenetic analysis showed that HVIV clustered within the clade of the Luckshill virus and is closely related to the family *Virgaviridae* (Appendix A). 

**(ix) *Closteroviruses***. The members of the family *Closteroviridae* have a mono-, bi- or tripartite positive-sense RNA genomes, which are 13 kb to 19 kb in size and non-enveloped [44]. One virus was detected in the weed *Alternanthera philoxeroides* in Hangzhou, with a genome length of 8823 bp. This virus shared 99.7% aa similarity with *Sedum sarmentosum crinivirus* (SSCV) (GenBank no. MN814305) and has a high abundance of 0.028 (Appendix A). SSCV was previously detected in plant, *Sedum sarmentosum* and was classified into the genus *Crinivirus* of the family *Closteroviridae* (Appendix A) [44]. SSCV-HZ was annotated as a variant of SSCV. The members of the genus *Crinivirus* could infect plants and cause diseases [45]. Such a close phylogenetic position suggests that SSCV-HZ may also infect plants and cause disease.

**(x) *Botourmiaviruses*, *Mitoviruses*, *Narnaviruses*, *Fusarivirus***. Mycoviruses detected in this study include the families, *Botourmiaviridae*, *Mitoviridae*, *Narnaviridae* and *Fusariviridae*. The members of the family *Botourmiaviridae* can infect plants and filamentous fungi, containing a mono- or multi-segmented positive-sense, and an ssRNA genome. The family *Botourmiaviridae* consists of four genera: *Ourmiavirus* (plant viruses), *Botoulivirus*, *Magoulivirus* and *Scleroulivirus* (mycoviruses). The members of the genera *Botoulivirus*, *Magoulivirus* and *Scleroulivirus* have positive-sense RNA genomes of 2 to 3 kb in size, while the genus *Ourmiavirus* has three segments of 2.8, 1.1 and 0.97 kb [46]. A total of 41 botourmiaviruses, including 35 novel viruses and 6 known viruses, were identified from 14 weed samples. The novel viruses were Guiyang botourmia-like virus (GBlV) 1 to 3 in Guiyang, Hangzhou botourmia-like virus (HBlV) 1 to 6 in Hangzhou, Sanya botourmia-like virus (SBolV) 1 to 21 in Sanya and Xiaogan botourmia-like virus (XBlV) 1 to 5 in Xiaogan. These botourmiaviruses have genome lengths ranging from 2061 to 3202 bp, and share 31.2% to 82.5% aa identities with the closest related viruses. In addition to these novel viruses, we also detected seven known viruses. They shared 91.8% to 98.0% aa identities with the already-described viruses Erysiphe necator-associated ourmia-like virus 2, Magnaporthe oryzae botourmiavirus 5, Magnaporthe oryzae botourmiavirus 6, Magnaporthe oryzae ourmia-like virus 4, Erysiphe necator-associated ourmia-like virus 97 and Plasmopara viticola lesion-associated ourmia-like virus 50. Importantly, the six known viruses were previously identified from fungal pathogens, such as *Erysiphe necator*, *Magnaporthe oryzae* and *Plasmopara viticola*. *Magnaporthe oryzae* is the causal agent of rice blast, the most destructive disease of rice that causes great yield losses annually [47]. Phylogenetic analysis indicated that 7 viruses clustered within the genus *Botoulivirus*, 2 viruses clustered within the genus *Penoulivirus*, 11 viruses were grouped within the genus *Magoulivirus*, 12 viruses clustered within the genus *Scleroulivirus* and 3 viruses clustered within the genus *Ourmiavirus* (Appendix A). Further analysis showed that there were two obvious branches within the genus *Scleroulivirus* (group 1 and group 2). Interestingly, nine novel botourmiaviruses clustered within group 2 with Pyricularia oryzae ourmia-like virus 2.

Viruses in the family *Mitoviridae* are non-encapsidated and have small positive-sense ssRNA genomes that are 2.2 to 3.0 kb in size, with a single ORF encoding a viral RNA-dependent RNA polymerase (RdRp) [48]. Mitoviruses are the only viruses that can infect eukaryotic mitochondria [49]. A total of 26 novel mitoviruses were detected in eight weed samples. Of the identified mitoviruses, 20 viruses, Guiyang mito-like virus (GMlV) 1 to 19 and Guiyang Paspalum thunbergii mito-like virus 1 (GPtMV1) were detected in Guiyang, with genome lengths between 2238 and 4342 bp. Two viruses, Hangzhou mito-like virus (HMlV) 1 to 2, were detected in Hangzhou, and have genome lengths between 2441 and 2513 bp. Three viruses, Sanya mito-like virus (SMlV) 1 to 3, were detected in Sanya, and the three viruses have RNA genomes between 2189 and 2796 bp in size. One virus, Xiaogan mito-like virus 1 (XMlV1), was identified in Xiaogan, with a genome length of 2279 bp (Appendix A). All the novel mitoviruses in this study displayed 34.0% to 81.9% aa identities with the closely related viruses (Appendix A). GMlV1 was identified in seven different weed species in Guiyang; SMlV1 was detected in two different weed species in Sanya (Appendix A). In addition, GPtMV1 has a high abundance of 0.402, suggesting that *Paspalum thunbergii* is a possible host of GPtMV1. Phylogenetic analysis showed that these viruses fell within the family *Mitoviridae* (Appendix A).

The members of the family *Narnaviridae* contain the simplest genomes ranging from 2.3 kb to 3.6 kb in size and encoding only a single polypeptide that has an RNA-dependent RNA polymerase domain [50]. Here, we identified 14 viruses as new members of the family *Narnaviridae*, including Guiyang narna-like virus (GNIV) 1 to 4, Guiyang Paspalum thunbergii narna-like virus 1 (GPtNV1), Hangzhou narna-like virus (HNlV) and Sanya narna-like virus (SNlV) 1 to 8. All these novel viruses in the family *Narnaviridae* have genome lengths ranging from 1837 to 3356 bp and share 26.5% to 74.6% aa identities with the closely related viruses (Appendix A). GPtNV1 displayed a high abundance of 0.814, suggesting that *Paspalum thunbergii* was the possible host of GPtNV1. In addition to these novel viruses, we identified three variants of the already-described Plasmopara viticola lesion-associated narnavirus 2 (PVaN 2) (GenBank no. MN539819), Erysiphe necator-associated narnavirus 6 (EnNV6) (GenBank no. MN605419) and Plasmopara viticola lesion-associated narnavirus 15 (PVaN 15) (GenBank no. MN539832), respectively. PVaN 2-SY was annotated as a variant of PVaN 2, EnNV6-HZ was annotated as a variant of EnNV6 and PVaN 15-SY was annotated as a variant of PVaN 15. These variants shared 92.2% to 99.3% aa identities with the closely related viruses. Analysis of genome structures showed that all the novel viruses contain a conserved RdRp domain, with some containing a reversed ORF (Appendix A). The phylogenetic analysis showed the novel viruses were divided into two groups (group 1 and group 2), 10 narnaviruses were located in group 1, which was related to the genus *Narnavirus*, and 7 narnaviruses grouped into group 2 (Appendix A). Importantly, group 2 was clearly distinct from group 1 in the evolutionary relationship. Thus, group 2 may be a new genus of the family *Narnaviridae*.

The family *Fusariviridae* is a newly established viral family, which has not yet been included in ICTV [51]. The genome of viruses in the family *Fusariviridae* can range from 6 kb to more than 10 kb and contain one to four ORFs [51,52]. We identified Sanya fusari-like virus 1 (SFlV1) in weed *Leersia hexandra* in Sanya, with a genome size of 6647 bp, and this virus clustered within the family *Fusariviridae* (Appendix A). Phylogenetic analysis showed that SFlV1 formed a monophyletic clade with Pleospora typhicola fusarivirus 1, Erysiphe necator-associated fusarivirus 2, Setosphaeria turcica fusarivirus 1, Plasmopara viticola lesion-associated fusarivirus 3, Plasmopara viticola lesion-associated fusarivirus 1 and Erysiphe necator-associated fusarivirus 1, and they shared high sequence similarities (83.0% to 87.0%, aa identities) (Appendix A).

**(xi) RNA bacteriophages.** Bacteriophages have been integrated into one class (*Leviviricetes*) and two orders (*Norzivirales* and *Timlovirales*). The order *Norzivirales* includes four families: *Atkinsviridae*, *Duinviridae*, *Fiersviridae* and *Solspiviridae*. The order *Timlovirales* contains two families: *Blumeviridae* and *Steitzviridae*. They all belong to the phylum *Lenarviricota* [53]. In this study, 79 viruses were classified into the phylum *Leviviricetes* including 78 novel viruses and 1 known virus (Appendix A). These novel viruses have genome lengths between 2098 and 4340 bp and displayed 29.5% to 77.8% aa identities with the closely related viruses. We also detected one known virus, and it shared 91.2% aa identity with the virus ssRNA phage SRR6960799 10 (GenBank no. BK014060). The phylogenetic analysis showed that 3 novel viruses clustered within the family *Atkinsviridae*, 1 novel virus was classified into the family *Duinviridae*, 64 novel viruses clustered within the family *Fiersviridae* and 5 novel viruses clustered within the family *Solspiviridae*, while the other 6 new viruses belong to the family *Steitzviridae* (Appendix A). The genome lengths of these novel viruses ranged from 2098 to 4340 bp, and shared 29.5% to 77.8% aa identities with the closely related viruses (Appendix A). Importantly, of the 79 bacteriophages, 72 were detected in Sanya, accounting for 91% of the identified bacteriophages, and indicating that RNA bacteriophages showed obvious geographical preference. All the RNA bacteriophages were identified with low abundance (Appendix A), indicating that these bacteriophages may be derived from endophytic bacteria. Thus, endophytic bacteria may be rich in weeds in paddy fields in Sanya.

### 3.2. Characterization of Negative-Sense RNA Viruses

The order *Bunyavirales* was one of the largest groups of segmented negative-sense single-stranded RNA viruses, which include many pathogenic viruses that infect arthropods, protozoa, plants and animals [54]. In this study, 24 novel negative-sense RNA viruses were identified in eight weed samples in Sanya, Xiaogan and Guiyang. These were Guiyang bunya-like virus 1 (GBlV1), Sanya bunya-like virus (SBLV) 1 to 20, Xiaogan peribunya-like virus 1 (XPeV1), Sanya leish-like virus 1 (SLlV1) and Xiaogan phenui-like virus 1 (XPhV1), with genome lengths ranging from 2540 to 11,361 bp. These viruses encode conserved RdRp protein and displayed 27.5% to 72.1% sequence similarities in aa levels with the closely related viruses (Appendix A). It is notable that each of the viruses (SBLV1 to -8) was detected in at least two weed samples in Sanya (Appendix A); for example, SBLV1 was detected in the weeds *Leersia hexandra* and *Leptochloa chinensis*. A large number of negative-sense RNA viruses in this study were identified from weeds in Sanya, and 71% of the viruses were found in weed *Leptochloa chinensis* in Sanya. Of the identified viruses, only three clustered within known viral families (Figure 5): XPeV1 was assigned into the family *Peribunyaviridae*, XPhV1 clustered within the family *Phenuiviridae* and SLlV1 clustered within the family *Leishbuviridae*. We were unable to classify the other 21 viruses into the present families; these viruses exhibited limited similarity (≤72.1%) to any well-defined family in the order *Bunyavirales*, and they only contained the L segment encoding RdRp (Figure 5 and Appendix A, Appendix A). In particular, the 21 newly identified viruses occupy topological positions belonging to two families, thus filling major phylogenetic gaps. Hence, we deemed that the branches may represent two new families of the *Bunyavirales* order. We named these branches *Alphabunyaviridae* and *Betabunyaviridae* (Figure 5).

### 3.3. Characterization of Double-Stranded RNA Viruses

The family *Partitiviridae* includes five genera, *Alphapartitivirus*, *Betapartitivirus*, *Gammapartitivirus*, *Deltapartitivirus* and *Cryspovirus* [55]. The members of the family *Partitiviridae* comprise non-enveloped viruses with bisegmented double-stranded (ds) RNA genomes that are 3.0 to 4.8 kb in size [55]. Here, we identified four dsRNA viruses in the weeds *Sagittaria trifolia*, *Digitaria sanguinalis*, *Paspalum distichum* and *Lindernia procumbens*; all the identified viruses clustered within the family *Partitiviridae*. Three novel viruses, Guiyang partiti-like virus 1 (GPlV1), Xiaogan partiti-like virus 1 (XPlV1) and Hangzhou partiti-like virus 1 (HPlV1), have genome lengths between 1689 and 1901 bp. These novel viruses displayed 59.2% to 77.9% aa identities with the closely related viruses. In addition, we identified one virus that displayed high sequence similarity (94.5%, aa identity) with the known virus Magnaporthe grisea partitivirus 1 (GenBank no. MK279458); it was annotated as a variant of Magnaporthe grisea partitivirus 1. The genomes of the members of the family *Partitiviridae* contain two segments: one encodes RdRp and the other encodes capsid protein [55]. The capsid-encoded segment was identified in Magnaporthe grisea partitivirus 1 and XPlV1, while it was not detected in GPlV1 and HPlV1 (Appendix A). Phylogenetic analysis revealed that XPlV1 clustered strongly within the genus *Alphapartitivirus*, suggesting that it was a new member of the genus *Alphapartitivirus.* In addition, GPlV1 formed a well-supported monophyletic group with maize-associated partiti-like virus; the new group appeared to represent a separate evolution distant from the members of the genus *Deltapartitivirus* and the genus *Cryspovirus*. Hangzhou partiti-like virus 1 was closely related to the genus *Gammapartitivirus*, suggesting that it may be a new member of the genus *Gammapartitivirus* (Figure 6).

## 4. Discussion

Here, we detected a diverse set of potential plant viruses from 29 weed samples by deep transcriptomic sequencing. We identified 12 complete or nearly complete genome sequences of the potential plant viruses, 11 of which are previously undescribed, and all the potential plant viruses are positive-sense RNA viruses (Appendix A). We also detected one known viruses, Sedum sarmentosum crinivirus. It was notable that this virus was firstly discovered in weeds. Of the potential plant viruses, most of them were detected in weeds at high abundance, indicating that weeds are the main hosts of these viruses. However, rice virus was not detected from the weed samples. Potyvirus and geminivirus, the most abundant plant virus species, were not identified in the collected weed samples [56,57]. Similar results were also obtained in tomato fields. Broad bean wilt virus 1 was found only in nightshade when there are numerous indications that this virus should be able to infect tomato [4]. The expected transfer is not observed for viral exchange between crops and weed plants, likely as a consequence of unforeseen biological or ecological barriers. Metatranscriptomics approaches have been successfully implemented to discover highly diverse and novel viruses from rice, citrus, sunflower and pear samples [14,58,59,60,61]. To the best of our knowledge, this is the first comprehensive high-throughput survey of viral sequences associated with weeds. 

Bacteriophages are the most abundant and diverse biological entities on the planet; they can even outnumber bacteria by approximately ten-fold in some ecosystems [62]. Bacteriophages can modulate the composition of microbial communities and maintain high levels of diversity, and they play important roles in the evolution, ecology and functions of microbial communities [63,64,65]. Bacteria have been described in a wide range of crop plant species [66,67]. For examples, indica and japonica varieties recruit large numbers of bacteria in rice fields [68]. Bacteriophage in Nicotiana benthamiana plants produce endolysin, which exhibits antimicrobial activity [69] Bacteriophage ACPWH in melon plants inhibits the progression of bacterial fruit botch [70]. In addition, bacteriophages play an important role in regulating the diversity of bacterial populations in plants [71]. In this study, we discovered 79 RNA bacteriophages in 29 different weeds, and our results show that 41% of the collected weeds harbor RNA bacteriophages, indicating that RNA bacteriophages may be common in weeds. However, the role of RNA bacteriophages in weed plants remains unknown. More studies are necessary to determine the potential functional roles of RNA bacteriophages in weeds.

Fungi harbor various viruses and interact with different plants. Viruses could infect fungi at the incidence rates from 3% to over 90%, and multiple viruses often coinfect fungal hosts, occasionally altering their phenotypes [72]. Fungi can inhabit plants in many different ways. Plants and fungi can be symbiotic or mutualistic. Endophytic fungi have been widely reported for their ability to aid in the defense of their host plants, and plant responses upon endophytic fungi colonization are also good for the immune system of the plant [73]. However, fungi can also be pathogenic to their plant hosts [74]. To date, there are few reports of mycoviruses being associated with weeds. In this study, our results showed that 85 viruses were clustered within the families of fungal virus species. We speculate that these mycoviruses may be derived from the fungi inhabiting weeds; however, we cannot exclude mycoviruses directly infecting weeds, especially those with high abundances. The relationship between mycoviruses and weeds remains relatively unexplored. More studies are necessary to survey the potential roles of mycoviruses in weeds.

In the present study, we described the metatranscriptomic sequencing of 29 weed samples. We discovered 304 virus genomes belonging to 224 virus species, including bacteriophages, mycoviruses and plant viruses. Of the identified viruses, 211 were newly discovered virus species and 13 were known virus species. The newly identified viruses clustered within 18 single-stranded positive-stranded RNA virus families, 3 single-stranded negative-stranded RNA virus families and 1 double-stranded RNA virus family. Many newly identified viruses constitute potentially new viral families and genera. Compared to previous results, our analysis of deep meta-transcriptomic sequencing has resulted in a great increase in the number of RNA viruses in weeds, with some RNA viruses clustering within the genera of several plant virus species. Together, our meta-transcriptomic analysis reveals genetic diversity of RNA viruses in weeds. Our work also provides valuable information for developing efficient strategies to manage weeds.

## Figures and Tables

**Figure 1 viruses-14-02489-f001:**
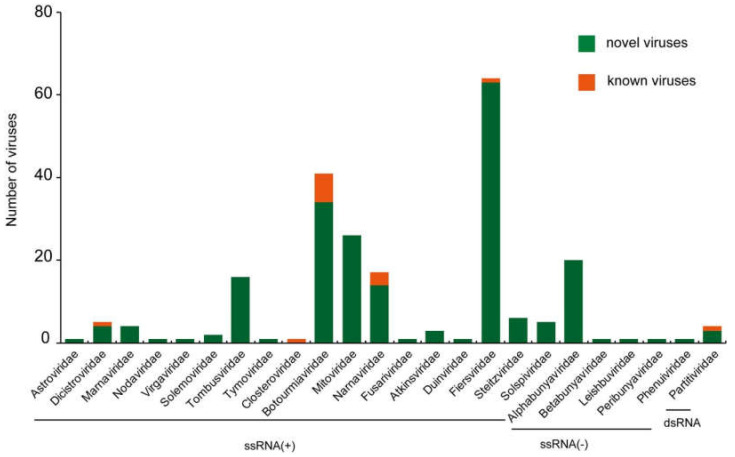
Overview of viral species identified in this study. According to the Baltimore classification, viruses identified in this study are classified into ssRNA(+), ssRNA(-) and dsRNA. The name of the classification is under the bar.

**Figure 2 viruses-14-02489-f002:**
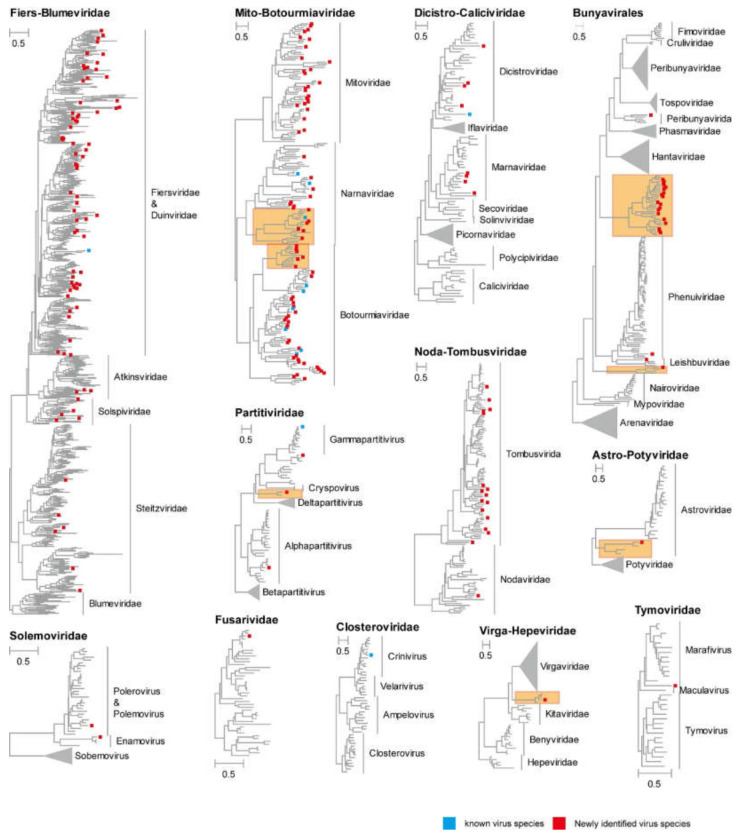
Phylogenetic diversity of RNA viruses identified in this study. Phylogenetic trees were constructed using the aligned sequences of RdRp. The name of each tree is shown to the top left of each phylogeny. Within each tree, the names of new viruses discovered in this study are marked with a red square, the names of known viruses discovered in this study are marked with a blue square and the families and genera of viruses (and branches in minimized trees) obtained in this study are marked in bold. Potential new families or genera are indicated with orange box.

**Figure 3 viruses-14-02489-f003:**
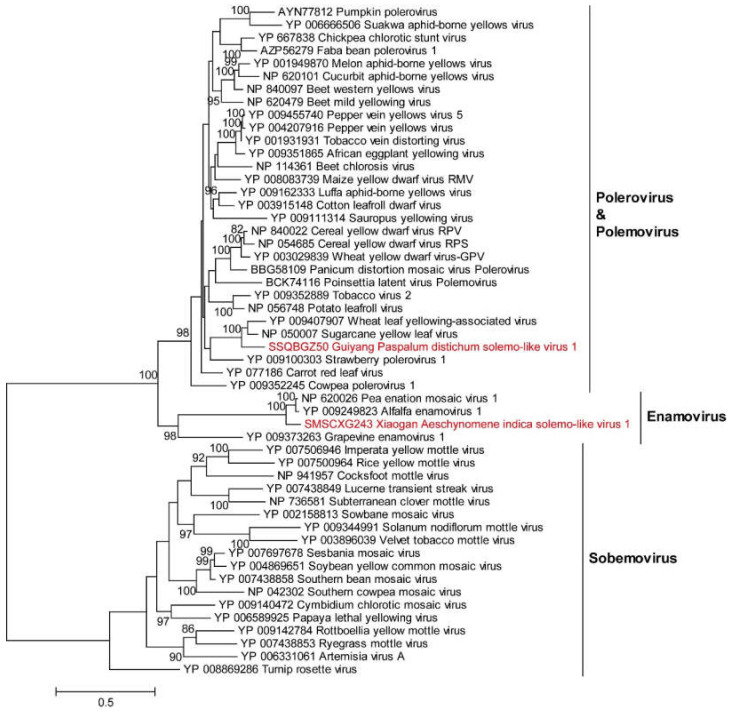
ML phylogenetic analysis of the family *Solemoviridae.* Phylogenetic trees were constructed using the aligned sequences of RdRp. The names of new viruses discovered in this study are marked in red, the reference virus serial numbers and names are marked in black and the families and genera of viruses (and branches in minimized trees) obtained in this study are marked in bold.

**Figure 4 viruses-14-02489-f004:**
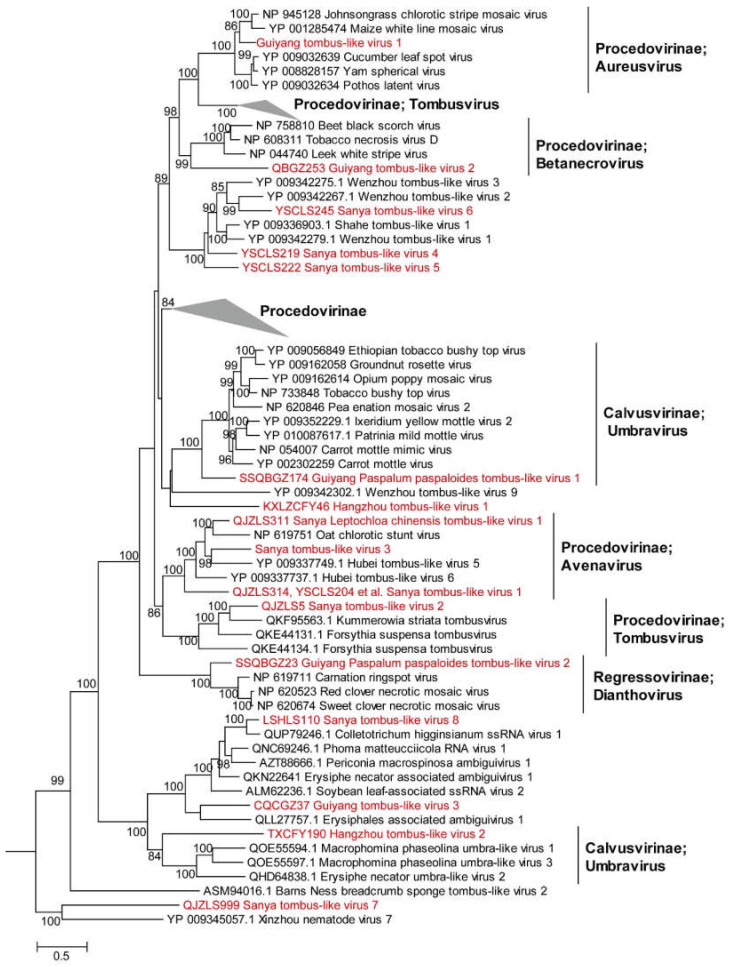
ML phylogenetic analysis of the family *Tombusviridae.* Phylogenetic trees were constructed using the aligned sequences of RdRp. The names of new viruses discovered in this study are marked in red, the reference virus serial numbers and names are marked in black and the families and genera of viruses (and branches in minimized trees) obtained in this study are marked in bold. The grey triangle means that the clade is all known viruses. The viruses were found in this study are not included in this clade. This clade is large, so it’s hidden.

**Figure 5 viruses-14-02489-f005:**
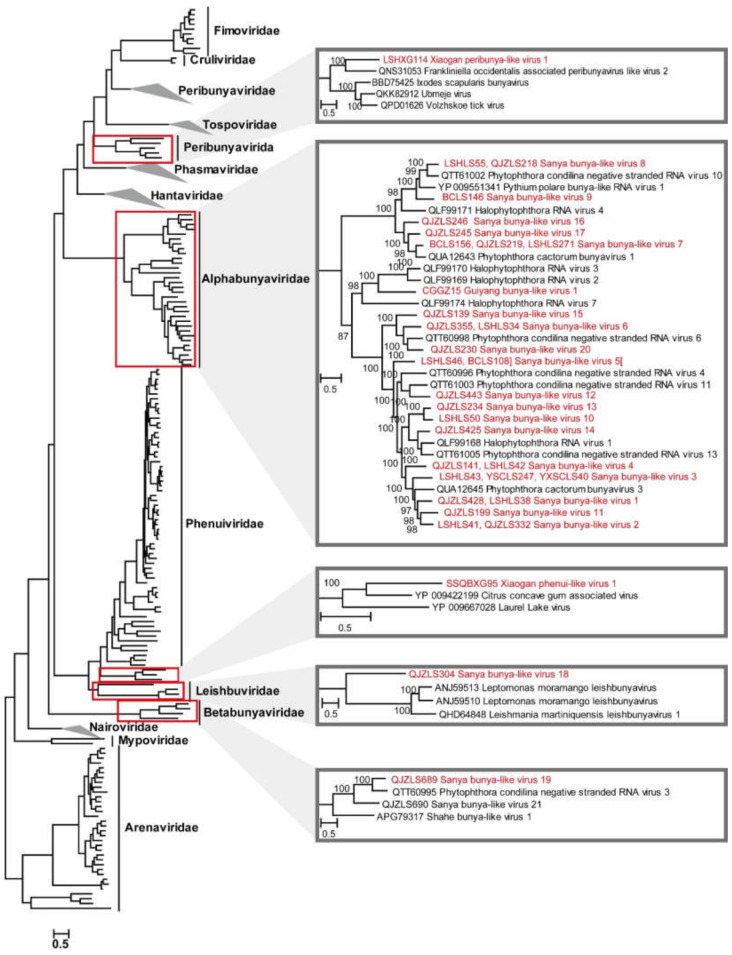
ML phylogenetic analysis of the order *Bunyavirales.* Phylogenetic trees were constructed using the aligned sequences of RdRp. The names of new viruses discovered in this study are marked in red, the names of known viruses discovered in this study are marked in blue, the reference virus serial numbers and names are marked in black and the families and genera of viruses (and branches in minimized trees) in this study are marked in bold. The red box represents the clade of the virus identified in this study. The grey triangle represents the enlarged part of the red box.

**Figure 6 viruses-14-02489-f006:**
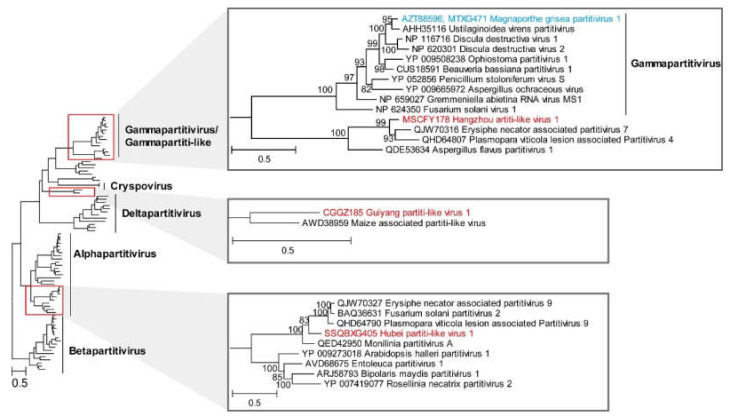
ML phylogenetic analysis of the family *Partitiviridae*. Phylogenetic trees were constructed using the aligned sequences of RdRp. The names of new viruses discovered in this study are marked in red, the names of known viruses discovered in this study are marked in blue, the reference virus serial numbers and names are marked in black and the genera of viruses (and branches in minimized trees) in this study are marked in bold. The red box represents the clade of the virus identified in this study. The grey triangle represents the enlarged part of the red box.

## Data Availability

Raw sequence data reads have been deposited in the NCBI Sequence Read Archive (SRA) under accession number SRR13060762-SRR13060768, SRR13060770- SRR13060790 and SRR13060792 (BioProject accession number PRJNA670260). All viral genome sequences generated in this study have been deposited in the GenBank database under accession numbers (OM514353-OM514623).

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
