# Peer review of "Novel RNA Viruses Discovered in Weeds in Rice Fields"

_viruses, 2022, doi:10.3390/v14112489_

Round 1

Reviewer 1 Report

Congratulations on the discovery of new RNA viruses! I have recommended that your manuscript be accepted for publication. Please address the following concerns.

Major revisions:

1.     The author's demonstrated the metatranscriptomic data which only detected RNA viruses and what about DNA viruses, is not even a single DNA virus detected from the weeds?

2.     Line 101: to provide additional details about the primers and program used for assembled sequences validated by RT-PCR.

3.     Line 149:  The authors have identified the 304 RdRP regions of viruses. If authors detected other parts of the viral genome by metatranscriptomic. Also include those results in that section which support additional validation of metatranscriptomics data.

4.     The authors should include more information about the other viruses other than, rice-affected viruses on weeds in the discussion part.

Minor corrections:

1.       Line 67: RNA extraction was performed by Trizol regent, clarify whether the extracted RNA was treated with DNase or not if treated mentioned it.

2.       Mention which part of the samples (leaf, stem, flower etc…) was used for RNA library creation.

3.       Line 102: provide the RACE kit details.

Author Response

Response to Reviewer 1 Comments

We would like to thank your insightful comments, which have helped us in redrafting the paper.

Major revisions:

  1. The author's demonstrated the metatranscriptomic data which only detected RNA viruses and what about DNA viruses, is not even a single DNA virus detected from the weeds?

In this study, our work focused on RNA viruses. Although some transcripts of DNA virus genes were detected in weed samples by metatranscriptomic sequencing, we could not assembly complete viral genomes. Currently we are trying to discover DNA viruses by Pacbac sequencing and illumina sequencing, and DNA viruses are identified as another project in our lab.

  1. Line 101: to provide additional details about the primers and program used for assembled sequences validated by RT-PCR.

We have listed the primers for RT-PCR in the table, named “The primers used for assembled sequences by RT-PCR”. To validate viral sequences, we firstly used Poly(A) Tailing Kit (Applied Biosystems, America) to add poly(A) to the 3’ end of the total RNA, then used SuperScriptTM III Reverse Transcription (ThermoFisher) for reverse transcription. Finally, RT-PCR amplification was performed using PrimerSTAR Max DNA Polymerase (Takara). The amplified products were cloned into pGEM T-Easy vector (Promega, WI) to Sanger sequencing.

The content was added in lines 104-109 of the revised version.

  1. Line 149: The authors have identified the 304 RdRP regions of viruses. If authors detected other parts of the viral genome by metatranscriptomic. Also include those results in that section which support additional validation of metatranscriptomics data.

In this study, functional domains of the assembled viral genomes were characterized using the following methods: (i) for viruses with closely related relatives, the functional domains were predicted by homology with known viral proteins; (ii) for viruses without a homologue in a closely related virus, the functional domains with each ORF was predicted using a domain-based blast search against the conserved domain database (https://www.ncbi.nlm.nih.gov/Structure/cdd/wrpsb.cgi) (expected threshold of IE-5). Signal peptides were determined using the SignalP-5.0 web service (http://www.cbs.dtu.dk/services/SignalP/). Transmembrane helices in ORFs were predicted using TMHMM v.2.0.

For the viruses with multiple segments, the segments were identified by homology to the proteins of the related reference proteins. For the potential segments without homology, we checked: (i) the sequencing depth of the segments; (ii) the presence of conserved regulatory sequences in 5' and/or 3' non-coding regions of the genome; (iii) the segments were found in the same samples; and (iv) the phylogenetic positions of related proteins.

We have added information about other parts of the viral genome by metatranscriptomic in “Materials and methods” in line 115-129 of the revised version.

  1. The authors should include more information about the other viruses other than, rice-affected viruses on weeds in the discussion part.

Potyvirus and geminivirus, the most abundant plant virus species, were not identified in the collected weed samples. Similar results were also obtained in tomato fields. Broad bean wilt virus 1 was found only in nightshade when there are numerous indications that this virus should be able to infect tomato. The expected transfer is not observed for viral exchange between crops and weed plants, likely as a consequence of unforeseen biological or ecological barriers.

The content was added in lines 539-545 of the revised version.

Minor corrections:

  1. Line 67: RNA extraction was performed by Trizol regent, clarify whether the extracted RNA was treated with DNase or not if treated mentioned it.

The extracted RNA was treated with DNase.

DNAase was adde in line 69 of the revised version.

  1. Mention which part of the samples (leaf, stem, flower etc…) was used for RNA library creation.

The leaf and stem of the samples were used to construct RNA library.

The content was changed in line 68 of the revised version.

  1. Line 102: provide the RACE kit details.

We have changed “5’/3’ RACE kits (TaKaRa)” to “5’/3’-Full RACE Core Set with PrimeScript™ RTase (TaKaRa)” in line 111 of the revised version.

Reviewer 2 Report

This metatranscriptomic deep sequencing study was conducted to identify RNA viruses in 29 samples of 23 weed species collected from rice fields in four rice production regions in China. A total of 221 novel RNA viruses were identified, 39 of them being sufficiently divergent to comprise new families and genera. Several species could infect plants. A large number of bacteriophages and mycoviruses was identified (although their role in weeds remains unknown). The results are presented clearly and in details, and are discussed convincingly. This study enlarged our knowledge on plant virus diversity, even beyond that of weeds. My only concern deals with the statements on the fluxes between weeds and cultivated plants (see below)

Major remarks

 - One aim of the study was to investigate the fluxes between weeds and cultivated plants (in particular, their putative role as ‘bridge’ host of viruses). The authors have a deep knowledge of the rice viruses, thanks to one of their recent articles on this very subject. They found no relationships between viruses occurring in rice and those in weeds collected in rice fields. The authors conclude that “… either the weeds cannot be infected by rice virus, or that weeds infected by rice viruses were not collected”.

- To my mind, a different approach and further information are needed to refine this conclusion. Selecting weeds from a large range of families (but with very few gramineaceous) did maximise the virus diversity found. Yet, it minimised the chances of finding rice viruses (because of the rice-virus host range). In contrast, collecting weeds almost exclusively from graminaceous would have maximised the changes of finding rice viruses.

- No information is provided about the “health” status of the rice fields in which weeds had been collected. Were the fields virus-infected? What about the rice fields in the immediate surroundings? If the rice fields were not infected, there would have been little chance to identify fluxes between rice fields and neighbouring weeds.

- What are the other crops nearby the collected weeds? Were they infected? If so, is there evidence of fluxes between the weeds and the viruses of these other crops?

- May be the authors could comment the conclusion of a recent article (quoted) on the same topic “… our results also highlight situations where an expected transfer is not observed, likely as a consequence of unforeseen biological or ecological barriers” (Ma et al. 2020).

- It is sometimes difficult to decipher what level of divergence make a new species, a new genus, a new family. I wonder whether the authors could clarify this point, possibly by providing, when available, levels derived from molecular taxonomy and/or ICTV recommendations.

- The lack of potyviruses and geminivures, the most abundant plant virus species, should be commented.

- I coud not open table S1 and table S5.

Minor remarks

How many fields were surveyed to collect the 29 samples?

p2, line 55: offer

p3, 1ine 09: a homologue

p 5, line 178: The names

p 6, line235: “the four novel solemoviruses” whereas line 235 “Here, we identified two novel solemoviruses…”.

p 7, line 253: “the family Marnaviridae includes 7 genera”. Only 6 are listed.

P 14, line 506: “We also detected three known viruses, Sedum sarmentosum crinivus”. What are the other two?

 The initials of the authors in reference 20 are missing.

Author Response

Response to Reviewer 2 Comments

We would like to thank your insightful comments, which have helped us in redrafting the paper.

Major remarks

- One aim of the study was to investigate the fluxes between weeds and cultivated plants (in particular, their putative role as ‘bridge’ host of viruses). The authors have a deep knowledge of the rice viruses, thanks to one of their recent articles on this very subject. They found no relationships between viruses occurring in rice and those in weeds collected in rice fields. The authors conclude that “… either the weeds cannot be infected by rice virus, or that weeds infected by rice viruses were not collected”.

- To my mind, a different approach and further information are needed to refine this conclusion. Selecting weeds from a large range of families (but with very few gramineaceous) did maximise the virus diversity found. Yet, it minimised the chances of finding rice viruses (because of the rice-virus host range). In contrast, collecting weeds almost exclusively from graminaceous would have maximised the changes of finding rice viruses.

Yes, I agree with you, our weed samples from a large range of families did maximize the virus diversity. In this study, we collected 29 weed samples from 24 sampling locations from four regions (Sanya, Xiaogan, Hangzhou and Guiyang). In addition to weed species of the family Gramineae, we also collected many other important weed plants in rice fields. We used metatranscriptomic sequencing to investigate the viral diversity in different weed species.

Our results showed that the numbers of identified viruses were significantly different in the collected weed species. The weed Leptochloa chinensis (family Gramineae) harbors up to 99 viruses in Sanya, the weed Monochoria vaginalis (family Pontederiaceae) harbors 31 viruses, and the weed Cyperus difformis (family Cyperaceae) contains 30 viruses. Thus, our data indicated that other weed species are also rich in viral diversity.

In this work, the metatranscriptomic sequencing conducts a preliminary analysis of the diversity of RNA viruses in weeds in rice fields. In the next step, we will collect multiple samples of the families Gramineae, Pontederiaceae and Cyperaceae to maximize the changes of finding rice viruses, and further determine whether weeds could be a “bridge” for rice virus transmission.

- No information is provided about the “health” status of the rice fields in which weeds had been collected. Were the fields virus-infected? What about the rice fields in the immediate surroundings? If the rice fields were not infected, there would have been little chance to identify fluxes between rice fields and neighbouring weeds.

The collected weeds showed mild symptoms, such as yellow spot leaves and a little dwarf plant. We sampled weeds in rice fields where several rice plants showed symptoms of viral infection.

- What are the other crops nearby the collected weeds? Were they infected? If so, is there evidence of fluxes between the weeds and the viruses of these other crops?

Except for rice, there were no other crops nearby the collected weeds. We collected weeds in rice fields where several rice plants showed yellow spot leaves and a little dwarf plant, our analysis showed that no evidence of virus fluxes was detected between rice crop and weeds.

- May be the authors could comment the conclusion of a recent article (quoted) on the same topic “… our results also highlight situations where an expected transfer is not observed, likely as a consequence of unforeseen biological or ecological barriers” (Ma et al. 2020).

Similar results were also obtained in tomato fields. Broad bean wilt virus 1 was only found in nightshade when there are numerous indications that this virus should be able to infect tomato. The expected transfer is not observed for viral exchange between crops and weed plants, likely as a consequence of unforeseen biological or ecological barriers.

The description was added in lines 541-545 of the revised version.

- It is sometimes difficult to decipher what level of divergence make a new species, a new genus, a new family. I wonder whether the authors could clarify this point, possibly by providing, when available, levels derived from molecular taxonomy and/or ICTV recommendations.

About a new species: novel viral species are identified as those that have < 90% RdRp protein identity to previously described viruses (Buck et al. 2005, Wille et al. 2019).

About a new genus: according to genus demarcation criteria of ICTV, different families have different classification criteria, which can be determined according to host species, genome segment, genome structure and so on. For example,

(i) the genera in the family Solemoviridae are differentiated by phylogenies of individual ORFs or proteins. Structural comparisons of encoded proteins or virions may help to determine phylogenetic relationships.

(ii) the genus demarcation criteria within family Partitiviridae: â‘  characteristic hosts within each genus (either plants or fungi for genera Alphapartititivirus and Betapartitivirus, fungi for genus Gammapartitivirus, plants for genus Deltapartitivirus and protozoa only for genus Cryspovirus); â‘¡ genome segment and protein lengths within a characteristic range for each genus; â‘¢ < 24% RdRp amino acid sequence identity in pairwise comparisons of viruses from different genera; â‘£separate phylogenetic grouping of RdRp sequences from each genus (https://ictv.global/report/chapter/partitiviridae/partitiviridae).

(iii) while the criteria can be applied to the demarcation of genera within the family Nodaviridae are: ①biological properties (host range, vectors, mode of transmission); ②virion physical/physicochemical characteristics (virion sedimentation coefficient and buoyant density); ③ structural protein characteristics (electrophoretic mobilities of the CP precursor or its cleavage products); ④antigenic properties; ⑤genome molecular characteristics (in the absence of sequence information, the electrophoretic mobilities of the viral genomic); ⑥ phylogeny (sequence of the two genomic RNAs, and their predicted proteins) (https://ictv.global/report/chapter/nodaviridae/nodaviridae).

The evolutionary relationship of RdRp amino acid sequence and/or other protein is a common criterion for the classification of new genera. Viruses belonging to a new genus formed a monophyletic clade, which is different from the known viral clade in a family.

About a new family: In the evolutionary relationship, the family clade formed by new viruses or/and unclassified viruses is significantly different from those formed by known viruses in ICTV, these clades might be considered as a new family.

References:

Buck K.W., Esteban R., Hillman B.I. Narnaviridae. In: Fauquet C.M., Mayo M.A., Maniloff J., Desselberger U., Ball L.A., editors. Virus Taxonomy: Eighth Report of the International Committee on Taxonomy of Viruses. 1st ed. Elsevier Academic Press; San Diego, CA, USA: 2005.

Wille M, Shi M, Klaassen M, Hurt AC, Holmes EC. Virome heterogeneity and connectivity in waterfowl and shorebird communities. ISME J. 2019 Oct;13(10):2603-2616.

- The lack of potyviruses and geminivirus, the most abundant plant virus species, should be commented.

Potyvirus and geminivirus, the most abundant plant virus species, were not identified in the collected weed samples. Similar results were also obtained in tomato fields. Broad bean wilt virus 1 was only found in nightshade when there are numerous indications that this virus should be able to infect tomato. The expected transfer is not observed for viral exchange between crops and weed plants, likely as a consequence of unforeseen biological or ecological barriers.

We have added information about the other viruses in discussion in lines 539-545 the revised version.

- I coud not open table S1 and table S5.

We have uploaded new table S1 and table S5 in supplementary materials.

Minor remarks

P2, How many fields were surveyed to collect the 29 samples?

We selected six sampling fields for each region (regions contain Hangzhou, Guiyang, Sanya and Xiaogan). Totally, we obtained 29 samples from at least 24 sampling fields.

p2, line 55: offer

We have changed “offerr” to “offer” in line 56 of the revised version.

p3, line 109: a homologue

We have changed “a homologues” to “a homologue” in line 117 of the revised version.

p 5, line 178: The names

We have changed “the names” to “The names” in line 200 of the revised version.

p 6, line235: “the four novel solemoviruses” whereas line 235 “Here, we identified two novel solemoviruses…”.

We have changed “the four novel solemoviruses” to “the two novel solemoviruses” in line 257 of the revised version.

p 7, line 253: “the family Marnaviridae includes 7 genera”. Only 6 are listed.

We have added another genus “Kusarnavirus” in line 278 of the revised version.

P 14, line 506: “We also detected three known viruses, Sedum sarmentosum crinivus”. What are the other two?

We identified only one known plant virus (Sedum sarmentosum crinivus) in weed samples, so we change “We also detected three known viruses, Sedum sarmentosum crinivus” to “We also detected one known virus, Sedum sarmentosum crinivus” in lines 535-536 of the revised version..

The initials of the authors in reference 20 are missing.

We have changed reference 20 to “Katoh, K.; Standley, D. M., MAFFT multiple sequence alignment software version 7: improvements in performance and usability. Mol Biol Evol 2013, 30, 772-780”. (lines 652-653 of the revised version.)

Round 2

Reviewer 1 Report

Dear Editor,

Accept the MS in its current form.

Thank you 

Abdul